# OKAMI: Teaching Humanoid Robots Manipulation Skills through Single Video Imitation

**Jinhan Li**[1†]   **Yifeng Zhu**[1*]   **Yuqi Xie**[1,2*]   **Zhenyu Jiang**[1,2*]   **Mingyo Seo**[1]

**Georgios Pavlakos**[1]   **Yuke Zhu**[1,2]

UT Austin[1]    NVIDIA Research[2]

**Abstract:** We study the problem of teaching humanoid robots manipulation skills by imitating from single video demonstrations. We introduce OKAMI, a method that generates a manipulation plan from a single RGB-D video and derives a policy for execution. At the heart of our approach is object-aware retargeting, which enables the humanoid robot to mimic the human motions in an RGB-D video while adjusting to different object locations during deployment. OKAMI uses open-world vision models to identify task-relevant objects and retarget the body motions and hand poses separately. Our experiments show that OKAMI achieves strong generalizations across varying visual and spatial conditions, outperforming the state-of-the-art baseline on open-world imitation from observation. Furthermore, OKAMI rollout trajectories are leveraged to train closed-loop visuomotor policies, which achieve an average success rate of 79.2% without the need for labor-intensive teleoperation. More videos can be found on our website https://ut-austin-rpl.github.io/OKAMI/.

**Keywords:** Humanoid Manipulation, Imitation From Videos, Motion Retargeting

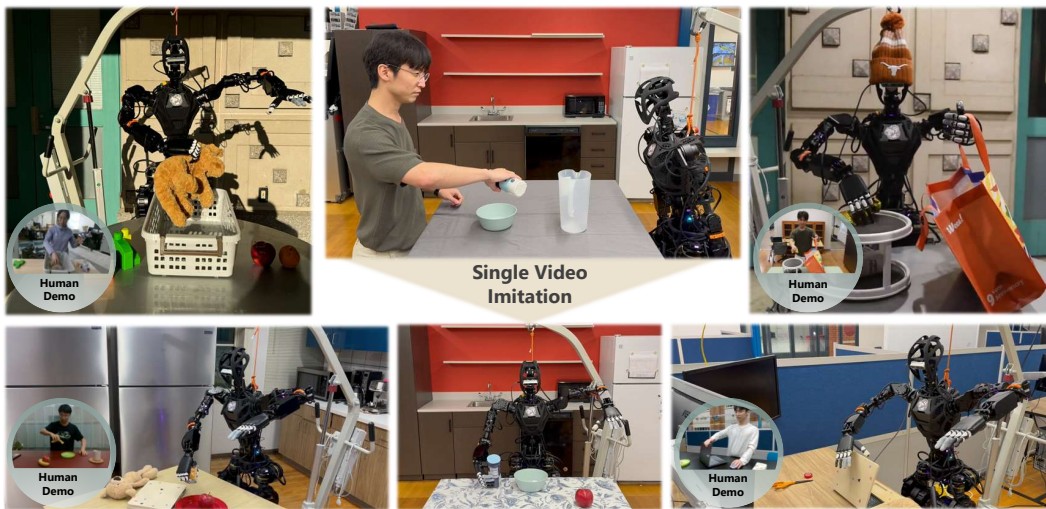

Figure 1: OKAMI enables a human user to teach the humanoid robot how to perform a new task by providing a single video demonstration.

## 1 Introduction

Deploying generalist robots to assist with everyday tasks requires them to operate autonomously in natural environments. With recent advances in hardware designs and increased commercial availability, humanoid robots emerge as a promising platform to deploy in our living and working spaces. Despite their great potential, they still struggle to operate autonomously and deploy robustly in the

---

[†] This work was done while Jinhan Li was a visiting researcher at UT Austin.

[*] Equal contribution.

8th Conference on Robot Learning (CoRL 2024), Munich, Germany.

unstructured world. A burgeoning line of work has resorted to deep imitation learning methods for humanoid manipulation [1–3]. However, they rely on large amounts of demonstrations through whole-body teleoperation, requiring domain expertise and strenuous efforts. In contrast, humans have the innate ability to watch their peers do a task once and mimic the behaviors. Equipping robots with the ability to imitate from visual observations will move us closer to the goal of training robotic foundation models from Internet-scale human activity videos.

We explore teaching humanoid robots to manipulate objects by watching humans. We consider a problem setting recently formulated as "open-world imitation from observation," where a robot imitates a manipulation skill from a single video of human demonstration [4–6]. This setting would facilitate users in effortlessly demonstrating tasks and enable a humanoid robot to acquire new skills quickly. Enabling humanoids to imitate from single videos presents a significant challenge — the video does not have action labels, but yet the robot has to learn to perform tasks in new situations beyond what's demonstrated in the video. Prior works on one-shot video learning have attempted to optimize robot actions to reconstruct the future object motion trajectories [4, 5]. However, they have been applied to single-arm manipulators and are computationally prohibitive for humanoid robots due to their high degrees of freedom and joint redundancy [7]. Meanwhile, the similar kinematic structure shared by humans and humanoids makes directly retargeting human motions to robots feasible [8, 9]. Nonetheless, existing retargeting techniques focus on free-space body motions [10–14], lacking the contextual awareness of objects and interactions needed for manipulation. To address this shortcoming, we introduce the concept of "object-aware retargeting". By incorporating object contextual information into the retargeting process, the resulting humanoid motions can be efficiently adapted to the locations of objects in open-ended environments.

To this end, we introduce OKAMI (**O**bject-aware **K**inematic ret**A**rgeting for hu**M**anoid **I**mitation), an object-aware retargeting method that enables a bimanual humanoid with two dexterous hands to imitate manipulation behaviors from a single RGB-D video demonstration. OKAMI uses a two-stage process to retarget the human motions to the humanoid robot to accomplish the task across varying initial conditions. The first stage processes the video to generate a reference manipulation plan. The second stage uses this plan to synthesize the humanoid motions through motion retargeting that adapts to the object locations in target environments.

OKAMI consists of two key designs. The first design is an open-world vision pipeline that identifies task-relevant objects, reconstructs human motions from the video, and localizes task-relevant objects during evaluation. Localizing objects at test time also enables motion retargeting to adapt to different backgrounds or new object instances of the same categories. The second design is the factorized process for retargeting, where we retarget the body motions and hand poses separately. We first retarget the body motions from the reference plan in the task space, and then warp the retargeted trajectory given the location of task-relevant objects. The trajectory of body joints is obtained through inverse kinematics. The joint angles of fingers are mapped from the plan onto the dexterous hands, reproducing hand-object interaction. With object-aware retargeting, OKAMI policies systematically generalize across various spatial layouts of objects and scene clutters. Finally, we train visuomotor policies on the rollout trajectories from OKAMI through behavioral cloning to obtain vision-based manipulation skills.

We evaluate OKAMI on human video demonstrations of diverse tasks that cover rich object interactions, such as picking, placing, pushing, and pouring. We show that its object-aware retargeting achieves 71.7% task success rates averaged across all tasks and outperforms the ORION [4] baseline by 58.3%. We then train closed-loop visuomotor policies on the trajectories generated by OKAMI, achieving an average success rate of 79.2%. Our contributions of OKAMI are three-fold:

1. OKAMI enables a humanoid robot to mimic human behaviors from a single video for dexterous manipulation. Its object-aware retargeting process generates feasible motions of the humanoid robot while adapting the motions to target object locations at test time;

2. OKAMI uses vision foundation models [15, 16] to identify task-relevant objects without additional human inputs. Their common-sense reasoning ability helps recognize task-

relevant objects even if they are not directly in contact with other objects or the robot hands, allowing our method to imitate more diverse tasks than prior work;

3. We validate OKAMI's strong spatial and visual generalization abilities on humanoid hardware. OKAMI enables real-robot deployment in natural environments with unseen object layouts, varying visual backgrounds, and new object instances.

## 2   Related Work

**Humanoid Robot Control.** Methods like motion planning and optimal control have been developed for humanoid locomotion and manipulation [10, 12, 17]. These model-based approaches rely on precise physical modeling and expensive computation [11, 12, 18]. To mitigate the stringent requirements, researchers have explored policy training in simulation and sim-to-real transfer [10, 19]. However, these methods still require a significant amount of labor and expertise in designing simulation tasks and reward functions, limiting their successes to locomotion domains. In parallel to automated methods, a variety of human control mechanisms and devices have been developed for humanoid teleoperation using motion capture suits [9, 12, 20–24], telexistence cockpits [25–29], VR devices [1, 30, 31], or videos that track human bodies [17, 32]. While these systems can control the robots to generate diverse behaviors, they require real-time human input that poses significant cognitive and physical burdens. In contrast, OKAMI only requires single RGB-D human videos to teach the humanoid robot new skills, significantly reducing the human cost.

**Imitation Learning for Robot Manipulation.** Imitation Learning has significantly advanced vision-based robot manipulation with high sample efficiency [33–44]. Prior works have shown that robots can learn visuomotor policies to complete various tasks with just dozens of demonstrations, ranging from long-horizon manipulation [34–36] to dexterous manipulation [37–39]. However, collecting demonstrations often requires domain expertise and high costs, creating challenges to scale. Another line of work focuses on one-shot imitation learning [40–44], yet they demand excessive data collection for meta-training tasks. Recently, researchers have looked into a new problem setting of imitating from a single video demonstration [4–6], referred to as "open-world imitation from observation" [4]. Unlike prior works that abstract away embodiment motions due to kinematic differences between the robot and the human, we exploit embodiment motion information owing to the kinematic similarity between humans and humanoids. Specifically, we introduce *object-aware retargeting* that adapts human motions to humanoid robots.

**Motion Retargeting.** Motion retargeting has wide applications in computer graphics and 3D vision [8], where extensive literature studies how to adapt human motions to digital avatars [45–47]. This technique has been adopted in robotics for recreating human-like motions on humanoid or anthropomorphic robots through various retargeting methods, including optimization-based approaches [11, 12, 20, 48], geometric-based methods [49], and learning-based techniques [10, 13, 17]. However, in manipulation tasks, these retargeting methods have been used within teleoperation systems, lacking a vision pipeline for automatic adaptation to object locations. OKAMI integrates the retargeting process with open-world vision, endowing it with object awareness so that the robot can mimic human motions from video demonstrations and adapt to object locations at test time.

## 3   OKAMI

In this work, we introduce OKAMI, a two-staged method that tackles open-world imitation from observation for humanoid robots. OKAMI first generates a *reference plan* using the object locations and reconstructed human motions from a given RGB-D video. Then, it retargets the human motion trajectories to the humanoid robot while adapting the trajectories based on new locations of the objects. Figure 2 illustrates the whole pipeline.

**Problem Formulation**   We formulate a humanoid manipulation task as a discrete-time Markov Decision Process defined by a tuple: $M = (S, A, P, R, \gamma, \mu)$, where $S$ is the state space, $A$ is the action space, $P(\cdot|s, a)$ is the transition probability, $R(s)$ is the reward function, $\gamma \in [0, 1)$ is the

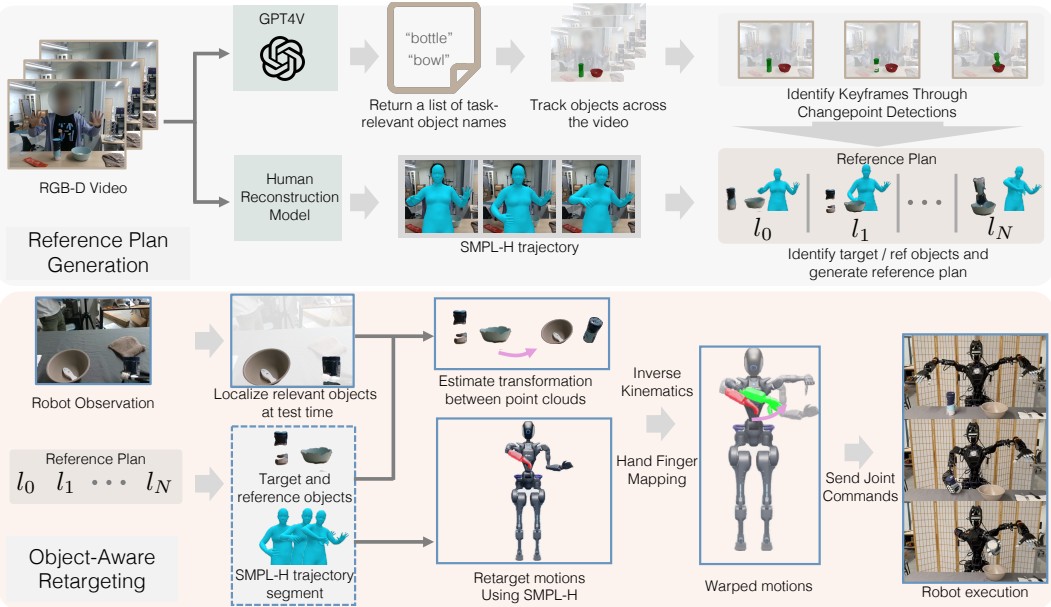

Figure 2: **Overview of OKAMI**. OKAMI is a two-staged method that enables a humanoid robot to imitate a manipulation task from a single human video. In the first stage, OKAMI generates a reference plan using GPT-4V and large vision models for subsequent manipulation. In the second stage, OKAMI follows the reference plan, where it retargets human motions onto the humanoid with object awareness. The retargeted motions are converted into a sequence of robot joint commands for the robot to follow.

discount factor, and $\mu$ is the initial state distribution. In our context, $S$ is the space of raw RGB-D observations that capture both the robot and object states, $A$ is the space of the motion commands for the humanoid robot, $R$ is the sparse reward function that returns 1 when a task is complete. The objective of solving a task is to find a policy $\pi$ that maximizes the expected task success rates from a wide range of initial configurations drawn from $\mu$ at test time.

We consider the setting of "open-world imitation from observation" [4], where the robot system takes a recorded RGB-D human video, $V$ as input, and returns a humanoid manipulation policy $\pi$ that completes the task as demonstrated in $V$. This setting is "open-world" as the robot does not have prior knowledge or ground-truth access to the categories or physical states of objects involved in the task, and it is "from observation" in the sense that video $V$ does not come with any ground-truth robot actions. A policy execution is considered successful if the state matches the state of the final frame from $V$. The success conditions of all tested tasks are described in Appendix B.1. Notably, two assumptions are made about $V$ in this paper: all the image frames in $V$ capture the human bodies, and the camera view of shooting $V$ is static throughout the recording.

### 3.1 Reference Plan Generation

To enable object-aware retargeting, OKAMI first generates a reference plan for the humanoid robot to follow. Plan generation involves understanding what task-relevant objects are and how humans manipulate them.

**Identifying and Localizing Task-Relevant Objects.** To imitate manipulation tasks from videos $V$, OKAMI must identify the task-relevant objects to interact with. While prior methods rely on unsupervised approaches with simple backgrounds or require additional human annotations [50–53], OKAMI uses an off-the-shelf Vision-Language Models (VLMs), GPT-4V, to identify task-relevant objects in $V$ by leveraging the commonsense knowledge internalized in the model. Concretely, OKAMI obtains the names of task-relevant objects by sampling RGB frames from the video demonstration $V$ and prompting GPT-4V with the concatenation of these images (details in Appendix A.2). Using these object names, OKAMI employs Grounded-SAM [16] to segment the objects in the first frame and track their locations throughout the video using a Vidoe Object

Segmentation model, Cutie [54]. This process enables OKAMI to localize task-relevant objects in $V$, forming the basis for subsequent steps.

**Reconstructing Human Motions.** To retarget human motions to the humanoid robot, OKAMI reconstructs human motions from $V$ to obtain motion trajectories. We adopt an improved version of SLAHMR [55], an iterative optimization algorithm that reconstructs human motion sequences. While SLAHMR assumes flat hands, our extension optimizes the hand poses of the SMPL-H model [56], which are initialized using estimated hand poses from HaMeR [57] (More details in Appendix A.1). This modification allows us to jointly optimize body and hand poses from monocular video. The output is a sequence of SMPL-H models capturing full-body and hand poses, enabling OKAMI to retarget human motions to humanoids (See Section 3.2). Additionally, the SMPL-H model can represent human poses across demographic differences, allowing easy mapping of motions from human demonstrators to the humanoid.

**Generating a Plan from Video.** Having identified task-relevant objects and reconstructed human motions, OKAMI generates a reference plan from $V$ for robots to complete each subgoal. OKAMI identifies subgoals by performing temporal segmentation on $V$ with the following procedure: We first track keypoints using CoTracker [58] and detect velocity changes of keypoints to determine keyframes, which correspond to subgoal states. For each subgoal, we identify a target object (in motion due to manipulation) and a reference object (serving as a spatial reference for the target object's movements through either contact or non-contact relations). The target object is determined based on the averaged keypoint velocities per object, while the reference object is identified through geometric heuristics or semantic relations predicted by GPT-4V (More implementation details of plan generation in Appendix A.4).

With subgoals and associated objects determined, we generate a reference plan $l_0, l_1, \ldots, l_N$, where each step $l_i$ corresponds to a keyframe and includes the point clouds of the target object $o_{\text{target}}$, the reference object $o_{\text{reference}}$, and the SMPL-H trajectory segment $\tau_{t_i:t_{i+1}}^{\text{SMPL}}$. If no reference object is required (e.g., grasping an object), $o_{\text{reference}}$ is null. Point clouds are obtained by back-projecting segmented objects from RGB images using depth images [59].

## 3.2 Object-Aware Retargeting

Given a reference plan from the video demonstration, OKAMI enables the humanoid robot to imitate the task in $V$. The robot follows each step $l_i$ in the plan by localizing task-relevant objects and retargeting the SMPL-H trajectory segment onto the humanoid. The retargeted trajectories are then converted into joint commands through inverse kinematics. This process repeats until all the steps are executed, and success is evaluated based on task-specific conditions (see Appendix B.1).

**Localizing Objects at Test Time.** To execute the plan in the test-time environment, OKAMI must localize the task-relevant objects in the robot's observations, extracting 3D point clouds to track object locations. By attending to task-relevant objects, OKAMI policies generalize across various visual conditions, including different backgrounds or the presence of novel instances of task-relevant objects.

**Retargeting Human Motions to the Humanoid.** The key aspect of *object-awareness* is adapting motions to new object locations. After localizing the objects, we employ a factorized retargeting process that synthesizes arm and hand motions separately. OKAMI first adapts the arm motions to the object locations so that the fingers of the hands are placed within the object-centric coordinate frame. Then OKAMI only needs to retarget fingers in the joint configuration to mimic how the demonstrator interacts with objects with their hands.

Concretely, we first map human body motions to the task space of the humanoid, scaling and adjusting trajectories to account for differences in size and proportion. OKAMI then warps the retargeted trajectory so that the robot's arm reaches the new object locations (More details in Appendix A.5). We consider two cases in trajectory warping — when the relational state between target and reference objects is unchanged and when it changes, adjusting the warping accordingly. In the first case,

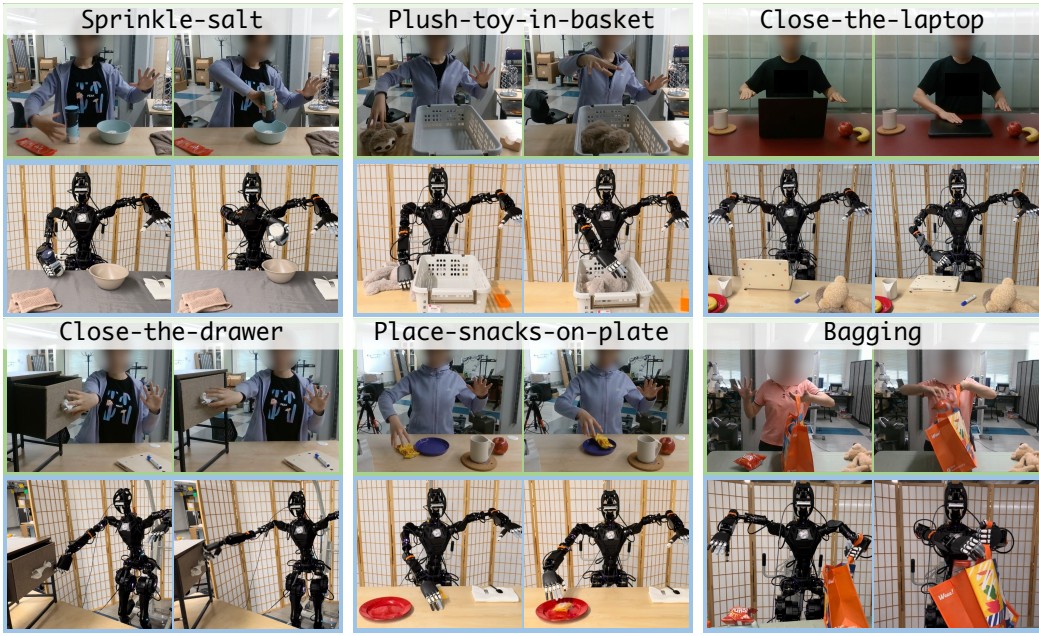

Figure 3: Visualization of initial and final frames of both human demonstrations and robot rollouts for all tasks.

we only warp the trajectory based on the target object locations; in the second case, the trajectory is warped based on the reference object location.

After warping, we use inverse kinematics to compute a sequence of joint configurations for the arms while balancing the weights of position and rotation targets in inverse kinematics computation to maintain natural postures. Simultaneously, we retarget the human hand poses to the robot's finger joints, allowing the robot to perform fine-grained manipulations (Implementation details in Appendix A.3). In the end, we obtain a full-body joint configuration trajectory for execution. Since arm motion retargeting is affine, our process naturally scales and adjusts motions from demonstrators with varied demographic characteristics. By adapting arm trajectories to object locations and retargeting hand poses independently, OKAMI achieves generalization across various spatial layouts.

## 4 Experiments

Our experiments are designed to answer the following research question: 1) Is OKAMI effective for a humanoid robot to imitate diverse manipulation tasks from single videos of human demonstration? 2) Is it critical in OKAMI to retarget the body motions of demonstrators to the humanoid robot instead of only retargeting based on object locations? 3) Can OKAMI retain its performances consistently on videos demonstrated by humans of diverse demographics? 4) Can the rollouts generated by OKAMI be used for training closed-loop visuomotor policies?

### 4.1 Experimental Setup

**Task Designs.** We describe the six tasks we use in the experiments: 1) `Plush-toy-in-basket`: placing a plush toy in the basket; 2) `Sprinkle-salt`: sprinkling a bit of salt into the bowl; 3) `Close-the-drawer`: pushing the drawer in to close it; 4) `Close-the-laptop`: closing the lid of the laptop; 5) `Place-snacks-on-plate`: placing a bag of snacks on the plate. 6) `Bagging`: placing a chip bag into a shopping bag. We select these six tasks that cover a diverse range of manipulation behaviors: `Plush-toy-in-basket` and `Place-snacks-on-plate` require pick-and-place behaviors of daily objects; `Sprinkle-salt` is the task that covers pouring behavior; `Close-the-drawer` and `Close-the-laptop` require the humanoid to interact with articulated objects, a prevalent type of interaction in daily environments; `Bagging` involves dexterous, bimanual manipulation and includes multiple subgoals. While we mainly focus on real

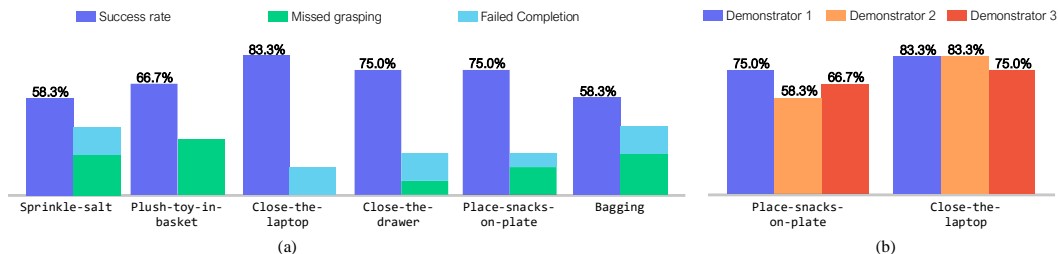

Figure 4: (a) Evaluation of OKAMI over all six tasks, including the success rates and the quantification of failed trials, separated by failure mode. (b) Evaluation of OKAMI using videos from different demonstrations. Demonstrator 1 is the main person recording videos for all evaluations in (a).

robot experiments, we also implement `Sprinkle-salt` and `Close-the-drawer` in simulation using RoboSuite [60] for easy reproducibility of OKAMI. See Appendix B.4.

**Hardware Setup.** We use a Fourier GR1 robot as our hardware platform, equipped with two 6-DoF Inspire dexterous hands and a D435i Intel RealSense camera for video recording and test-time observation. We implement a joint position controller that operates at 400Hz. To avoid jerky movements, we compute joint position commands at 40Hz and interpolate the commands to 400Hz trajectories.

**Evaluation Protocol.** We run 12 trials for each task. The locations of the objects are randomly initialized within the intersection of the robot camera's view and the humanoid arms' reachable range. The tasks are evaluated on a tabletop workspace with multiple objects, including both task-relevant objects and various other objects. Further, we test new object generalization on `Place-snacks-on-plate`, `Plush-toy-in-basket`, and `Sprinkle-salt` tasks, changing the involved plate, snack bag, plush toy, and bowl to other instances of the same type.

**Baselines.** We compare our result with a baseline ORION [4]. Since ORION was proposed for parallel-jaw grippers, it is not directly applicable in our experiments and we adopt it with minimal modifications: we estimate the palm trajectory using the SMPL-H trajectories, and warp the trajectory conditioning on the new object locations. The warped trajectory is used in the subsequent inverse kinematics for computing robot joint configurations.

### 4.2 Quantitative Results

To answer question (1), we evaluate the policies of OKAMI across all the tasks, covering diverse behaviors such as daily pick-place, pouring, and manipulation of articulated objects. The results are presented in Figure 4(a). In our experiment, we randomly initialize the object locations so that the robot needs to adapt to the locations of the objects. This result shows the effectiveness of OKAMI in generalizing over different visual and spatial conditions.

To answer question (2), we compare OKAMI against ORION on two representative tasks, `Place-snacks-on-plate` and `Close-the-laptop`. In the comparison experiment, OKAMI differs from ORION in that ORION does not condition on the human body poses. OKAMI achieves 75.0% and 83.3% success rates, respectively, while ORION only achieves 0.0% and 41.2%, respectively. Additionally, we compare OKAMI against ORION on the two simulated versions of `Sprinkle-salt` and `Close-the-drawer` tasks. In simulation, OKAMI achieves 82.0% and 84.0% success rates in two tasks while ORION only achieves 0.0% and 10.0%. Most failures of ORION policies are due to failing to approach objects with reliable grasping poses (e.g., in `Place-snacks-on-plate` task, ORION tries to grasp the snack from the sides instead of the top-down grasp in human video), and failing to rotate the wrist fully to achieve behaviors such as pouring. These behaviors originate from the fact that ORION ignores the embodiment information, thus falling short in performance compared to OKAMI. The superior performance of OKAMI suggests the importance of retargeting the body motion of the human demonstrators onto the humanoid when imitating from human videos.

To answer question (3), we conduct a controlled experiment of recording videos of different demonstrators and test if OKAMI policies maintain strong performance across the video inputs.

Same as the previous experiment, we evaluate OKAMI on the `Place-snacks-on-plate` and `Close-the-laptop` tasks. The results are presented in Figure 4(b). We show that for the task `Close-the-laptop`, there is no statistical significance in performance change. As for task `Place-snacks-on-plate`, while the evaluation maintains above 50%, the worst policy performance is 16.7% worse than the best policy performance. After looking into the video recording, we find that the motion of demonstrator 2 is relatively faster than the other two demonstrators, and faster motions create a noisy estimation of motion when doing human model reconstruction. Overall, OKAMI can maintain reasonably good performance given videos from different demonstrators, but there is room for improvements on our vision pipeline to handle such variety.

### 4.3 Learning Visuomotor Policy With OKAMI Rollout Data

We address question (4) by training neural visuomotor policies on OKAMI rollouts. We first run OKAMI over randomly initialized object layouts to generate multiple rollouts and collect a dataset of successful trajectories while discarding the failed ones. We train neural network policies on this dataset through a behavioral cloning algorithm. Since smooth execution is critical for humanoid manipulation, we implement the behavioral cloning with ACT [61], which predicts smooth actions via its temporal ensemble design, a trajectory smoothing component (more implementation details in Appendix B.5).

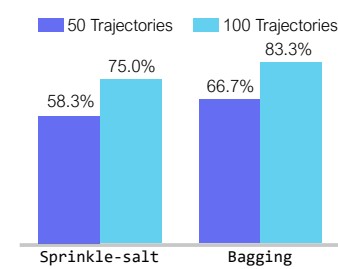

Figure 5: Success rates of learned visuomotor policies on `Sprinkle-salt` and `Bagging` using 50 and 100 trajectories, respectively.

We train visuomotor policies for `Sprinkle-salt` and `Bagging`. Figure 5 illustrates the success rates of these policies, demonstrating that OKAMI rollouts are effective data sources for training. We also show that the learned policies improve as more rollouts are collected. These results hold the promise of scaling up data collection for learning humanoid manipulation skills without laborious teleoperation.

## 5   Conclusion

This paper introduces OKAMI that enables a humanoid robot to imitate a single RGB-D human video demonstration. At the core of OKAMI is object-aware retargeting, which retargets the human motions onto the humanoid robot and adapts the motions to the target object locations. OKAMI consists of two stages to realize object-aware retargeting. The first stage is generating a reference plan for manipulation from the video. The second stage is used for retargeting, where OKAMI retargets the arm motions in the task space and the finger motions in the joint configuration space. Our experiments validate the design of OKAMI, showing the systematic generalization of OKAMI policies. OKAMI enables efficient collection of trajectory data based on a single human video demonstration. OKAMI-based data collection significantly reduces the human cost for policy training compared to that required by teleoperation.

**Limitations and Future Work.** The current focus of OKAMI is on the upper body motion retargeting of humanoid robots, particularly for manipulation tasks within tabletop workspaces. A promising future direction is to include lower body retargeting that enables locomotion behaviors during video imitation. To enable full-body loco-manipulation, a whole-body motion controller needs to be implemented as opposed to the joint position controller used in OKAMI. Additionally, we rely on RGB-D videos in OKAMI, which limits us from using in-the-wild Internet videos recorded in RGB. Extending OKAMI to use web videos will be another promising direction for future works. At last, the current implementation of retargeting has limited robustness against large variations in object shapes. A future improvement would be integrating more powerful foundation models that endow the robot with a general understanding of how to interact with a class of objects in spite of their large shape changes.

**Acknowledgments**

We would like to thank William Yue for providing the initial implementation of the behavioral cloning policies, Peter Stone for his valuable support with task designs and demo shooting, Yuzhe Qin for sharing the dex-retargeting codebase, and Zhenjia Xu for his advice on developing the humanoid robot infrastructure.

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

# A   Implementation Details

## A.1   Human Reconstruction From Videos

**Method.** For the 3D human reconstruction, we start by tracking the person in the video and getting an initial estimate of their 3D body pose using 4D Humans [62]. This body reconstruction cannot capture the hand pose details (i.e., the hands are flat). Therefore, for each detection of the person in the video, we detect the two hands using ViTPose [63], and for each hand, we apply HaMeR [57] to get an estimate of the 3D hand pose. However, the hands reconstructed by HaMeR can be inconsistent with the arms from the body reconstruction (e.g., different wrist orientation and location). To address this, we apply an optimization refinement to make the body and the hands consistent in each frame, and encourage that the holistic body and hands motion is smooth over time. This optimization is similar to SLAHMR [55], with the difference that besides the body pose and location of the SMPL+H model [56], we also optimize the hand poses. We initialize the procedure using the 3D body pose estimate from 4D Humans and the 3D hand poses from HaMeR. Moreover, we use the 2D projection of the 3D hands predicted by HaMeR to constrain the projection of the 3D hand keypoints of the holistic model using a reprojection loss. Finally, we can jointly optimize all the parameters (body location, body pose, hand poses) over the duration of the video, as described in SLAHMR [55].

Our modified SLAHMR incorporates the SMPL-H model [56] to include hand poses in the human motion reconstruction. We initialize hand poses in each frame using 3D hand estimates from HaMeR [57]. The optimization process then jointly refines body locations, body poses, and hand poses over the video sequence. This joint optimization allows for accurate modeling of how hands interact with objects, which is crucial for manipulation tasks.

The optimization minimizes the error between the 2D projections of the 3D joints from the SMPL-H model and the detected 2D joint locations from the video. We use standard parameters and settings as described in SLAHMR [55], adapting them to accommodate the SMPL-H model.

**Inference Requirements.** The model of human reconstruction we use is large and needs to be run on a computer with sufficiently good computation speed. Here we provide details about the runtime performance of the human reconstruction model. We use a desktop that comes with a GPU RTX3090 that has the size of the memory 24 GB. For a 10 seconds video with fps 30, it processes 10 minutes.

## A.2   Prompts of Using GPT4V

In order to use GPT4V in OKAMI, we need GPT4V's output to be in a typed format so that the rest of the programs can parse the result. Moreover, in order for the prompts to be general across a diverse set of tasks, our prompt does not leak any task information to the model. Here we describe the three different prompts in OKAMI for using GPT4V.

**Identify Task-relevant Objects.**   OKAMI uses the following prompt to invoke GPT4V so that it can identify the task-relevant objects from a provided human video:

**Prompt:** You need to analyze what the human is doing in the images, then tell me: 1. All the objects in front scene (mostly on the table). You should ignore the background objects. 2. The objects of interest. They should be a subset of your answer to the first question. They are likely the objects manipulated by human or near human. Note that there are irrelevant objects in the scene, such as objects that does not move at all. You should ignore the irelevant objects.

Your output format is:

```
The human is xxx.
All objects are xxx.
The objects of interest are:
```json
{
```

```
        "objects": ["OBJECT1", "OBJECT2", ...],
}
```

Ensure the response can be parsed by Python 'json.loads', e.g.: no trailing commas, no single quotes, etc. You should output the names of objects of interest in a list ["OBJECT1", "OB-JECT2", ...] that can be easily parsed by Python. The name is a string, e.g., "apple", "pen", "keyboard", etc.

**Identify Target Objects.** OKAMI uses the following prompt to identify the target object of each step in the reference plan:

**Prompt:** The following images shows a manipulation motion, where the human is manipulating an object.

Your task is to determine which object is being manipulated in the images below. You need to choose from the following objects: {a list of task-relevant objects}.

Tips: the manipulated object is the object that the human is interacting with, such as picking up, moving, or pressing, and it is in contact with the human's {the major moving arm in this step} hand.

Your output format is:

```json
{
        "manipulate_object_name": "MANIPULATE_OBJECT_NAME",
}
```

Ensure the response can be parsed by Python 'json.loads', e.g.: no trailing commas, no single quotes, etc.

**Identify Reference Objects.** Here is the prompt that asks GPT4V to identify the reference object of each step in the reference plan:

**Prompt:** The following images shows a manipulation motion, where the human is manipulating the object {manipulate_object_name}.

Please identify the reference object in the image below, which could be an object on which to place {manipulate_object_name}, or an object that {manipulate_object_name} is interacting with. Note that there may not necessarily have an reference object, as sometimes human may just playing with the object itself, like throwing it, or spinning it around. You need to first identify whether there is a reference object. If so, you need to output the reference object's name chosen from the following objects: {a list of task-relevant objects}.

Your output format is:

```json
{
        "reference_object_name": "REFERENCE_OBJECT_NAME" or "None",
}
```

Ensure the response can be parsed by Python 'json.loads', e.g.: no trailing commas, no single quotes, etc.

### A.3 Details on Factorized Process for Retargeting

**Body Motion Retarget.** To retarget body motions from the SMPL-H representation to the humanoid, we extract the shoulder, elbow, and wrist poses from the SMPL-H models. We then use inverse kinematics to solve the body joints on the humanoid, ensuring they produce similar shoulder and elbow orientations and similar wrist poses. The inverse kinematics is implemented using an open-sourced library Pink [64]. The IK weights we use for shoulder orientation, elbow orientation, wrist orientation, and wrist position are 0.04, 0.04, 0.08, and 1.0, respectively.

**Hand Pose Mapping.** As we describe in the method section, we first retarget the hands from SMPL-H models to the humanoid's dexterous hands using a hybrid implementation of inverse kinematics and angle mapping. Here are the details of how this mapping is performed. Once we obtain the SMPL-H models from a video demonstration, we can obtain the locations of 3D joints from the hand mesh models from SMPL-H. Subsequently, we can compute the rotating angles of each joint that correspond to certain hand poses. Then we apply the computed joint angles to the hand meshes of a canonical SMPL-H model, which is pre-defined to have the same size as the humanoid robot hardware. From this canonical SMPL-H model, we can get the 3D keypoints of hand joints and use an existing package, dex-retarget, an off-the-shelf optimization package to directly compute the hand joint angles of the robot [65].

**Inverse Kinematics.** After warping the arm trajectory, we use inverse kinematics to compute the robot's joint configurations. We assign weights of 1.0 to hand position and 0.08 to hand rotation, prioritizing accurate hand placement while allowing the arms to maintain natural postures.

For retargeting human hand poses to the robot, we map the human hand joint angles to the corresponding joints in the robot's hand. This enables the robot to replicate fine-grained manipulations demonstrated by the human, such as grasping and object interaction. Our implementation ensures that the retargeted motions are physically feasible for the robot and that overall execution appears natural and effective for the task at hand.

### A.4 Additional Details of Plan Generation

For temporal segmentation, we sample keypoints from the segmented objects in the first frame and track them across the video using CoTracker [58]. We compute the average velocity of these keypoints at each frame and apply an unsupervised changepoint detection algorithm [66] to detect significant changes in motion, identifying keyframes that correspond to subgoal states.

To determine contact between objects, we compute the relative spatial locations and distances between the point clouds of objects. If the distance between objects falls below a predefined threshold, we consider them to be in contact. For non-contact relations that are difficult to infer geometrically—such as a cup in a pouring task—we use GPT4V to predict semantic relations based on the visual context. GPT4V can infer that the cup is the recipient in a pouring action even if there is no direct contact.

### A.5 Trajectory Warping

Here, we mathematically describe the process of trajectory warping. We denote the trajectory for robot as $\tau^{\text{robot}}$ retargeted from $\tau^{\text{SMPL}}_{t_i:t_{i+1}}$ in the generated plan. Denote the starting point and end point of $\tau^{\text{robot}}$ as $p_{\text{start}}, p_{\text{end}}$, respectively. Note that all points along the trajectory are represented in SE(3) space.

Each point $p_t$ on the original retargetd trajectory can be described by the following function:

$$p_t = p_{\text{start}} + (\tau^{\text{robot}}(t) - p_{\text{start}}) \tag{1}$$

where $t \in \{t_i, \ldots, t_{i+1}\}$, $\tau^{\text{robot}}(t_i) = p_{\text{start}}$, $\tau^{\text{robot}}(t_{i+1}) = p_{\text{end}}$.

When warping the trajectory, we either only needs to adapt the trajectory to the new target object location, or adapt the trajectory to the new locations of both the target and the reference objects,

as described in Section 3.2. Without loss of generality, we denote the SE(3) transformation for the starting point is $T_{\text{start}}$, and the SE(3) transformation for the end point is $T_{\text{end}}$. Now the warped trajectory can be described by the following function:

$$p_t = T_{\text{start}} \cdot p_{\text{start}} + (\hat{\tau}^{\text{robot}}(t) - T_{start} \cdot p_{\text{start}}) \tag{2}$$

where $\hat{\tau}^{\text{robot}}(t) = \frac{\tau^{\text{robot}}(t) - p_{\text{start}}}{p_{\text{end}} - p_{\text{start}}}(T_{\text{end}} \cdot p_{\text{end}} - T_{\text{start}} \cdot p_{\text{start}}) + T_{\text{start}} \cdot p_{\text{start}}, \forall t \in \{t_i, \ldots, t_{i+1}\}$. In this way, we have $\hat{\tau}^{\text{robot}}(t_i) = T_{\text{start}} \cdot p_{\text{start}}, \hat{\tau}^{\text{robot}}(t_{i+1}) = T_{\text{end}} \cdot p_{\text{end}}$. Note that this trajectory warping assumes the end point of a trajectory is not the same as the starting point, which is a common assumption for most of the manipulation behaviors.

# B  Additional Experimental Details

## B.1  Success Conditions

We describe the success conditions we use to evaluate if a task rollout is successful or not.

- `Sprinkle-salt`: The salt bottle reaches a position where the salt is poured out into the bowl.
- `Plush-toy-in-basket`: The plush toy is put inside the container, with more than 50% of the toy inside the container.
- `Close-the-laptop`: The display is lowered towards the base until the two parts meet at the hinge (aka the laptop is closed).
- `Close-the-drawer`: The drawer is pushed back to the containing region, either it's a drawer or a layer of a cabinet.
- `Place-snacks-on-plate`: The snack is placed on top of the plate, with more than 50% of the snack package on the plate.
- `Bagging`: The chip bag is put into the shopping bag which is initially closed.

## B.2  Implementation of Baseline

We implement the baseline ORION [4] with minimal modifications to apply it to our humanoid setting. First, we estimate the palm trajectory from SMPL-H trajectories by using the center point of the reconstructed fingers as the palm position at each time step. Next, we warp the palm trajectory based on the test-time objects' locations. Finally, we use inverse kinematics to solve for the robot's body joints, with the warped trajectory serving as the target palm position.

## B.3  Details on Different Demonstrators

Figure 6 shows the videos of three different human demonstrators performing `Place-snacks-on-plate` and `Close-the-laptop` tasks. We calculate the success rates of imitating different videos, and the results are shown in Figure 4(b).

## B.4  Simulation Evaluation

For easy reproducibility, we replicate two tasks, `Sprinkle-salt` and `Close-the-drawer`, in simulation (Figure 7). We implement these tasks using robosuite [60], which recently provided cross-embodiment support, including humanoid manipulation. We use "GR1FixedLowerBody" as the robot embodiment in these two tasks.

Note that for the policy of each task, we use the same human video as the ones used in real robot experiments. We compare three methods in simulation: OKAMI (w/vision), OKAMI (w/o vision), and ORION. OKAMI (w/vision) the same method we use in our real robot experiments. OKAMI (w/o vision) is the simplified version of OKAMI where we assume the model directly gets the

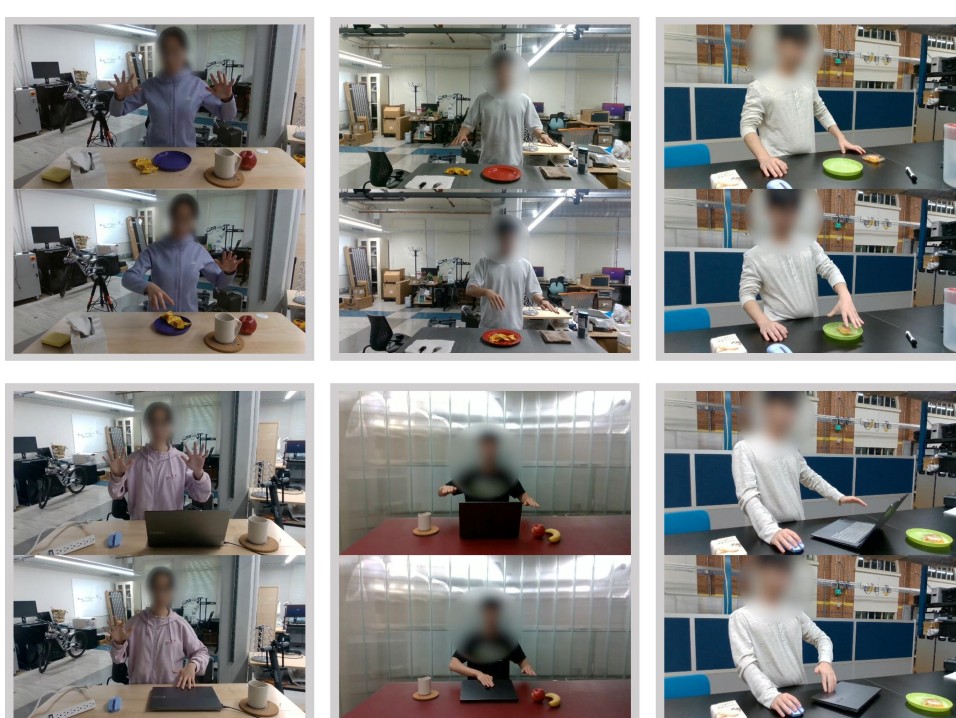

Figure 6: The initial and end frames of videos performed by different human demonstrators. The first row is `Place-snacks-on-plate` task, and the second row is `Close-the-laptop` task.

ground-truth poses of objects. The evaluation results are shown in Table 1, where each reported number is the success rate averaged over 50 rollouts.

We notice that the simulation results are generally better than the real robot experiments. The performance difference comes from the easy physical interaction between dexterous hands and objects compared to the real robot hardware. Also, OKAMI without vision can achieve a much higher success rate than OKAMI with vision because the noise and uncertainty of perception are abstracted away. Specifically, a large portion of uncertainties come from the partial observation of object point clouds, and the estimation of the object location is off the ground-truth locations of objects, while the success of OKAMI highly depends on the quality of trajectory warping, which is dependent on the correct estimation of object locations. This simulation result also indicates that the performance of OKAMI is expected to improve if more powerful vision models with higher accuracy are available.

| Method | Sprinkle-salt | Close-the-drawer |
|---|---|---|
| OKAMI (w/ vision) | 82% | 84% |
| OKAMI (w/o vision) | 100% | 100% |
| ORION | 0% | 10% |

Table 1: The average success rates (%) across different methods in two tasks, `Sprinkle-salt` and `Close-the-drawer`

## B.5 Visuomotor Policy Details

We choose ACT [61] in our experiments for behavioral cloning, an algorithm that has been shown effective in learning humanoid manipulation policies [67]. Notably, we choose pretrained DinoV2 [68, 69] as the visual backbone of a policy. The policy takes a single RGB image and 26-dimension joint positions as input and outputs the action of the 26-dimension absolute joint position for the robot to reach. In Table 2, we show the hyperparameters used for behavioral cloning.

| | |
|---|---|
| KL weight | 10 |
| chunk size | 60 |
| hidden dimension | 512 |
| batch size | 45 |
| feedforward dimension | 3200 |
| epochs | 25000 |
| learning rate | 5e-5 |
| temporal weighting | 0.01 |

Table 2: The hyperparameters used in ACT.

Close-the-drawer        Sprinkle-salt

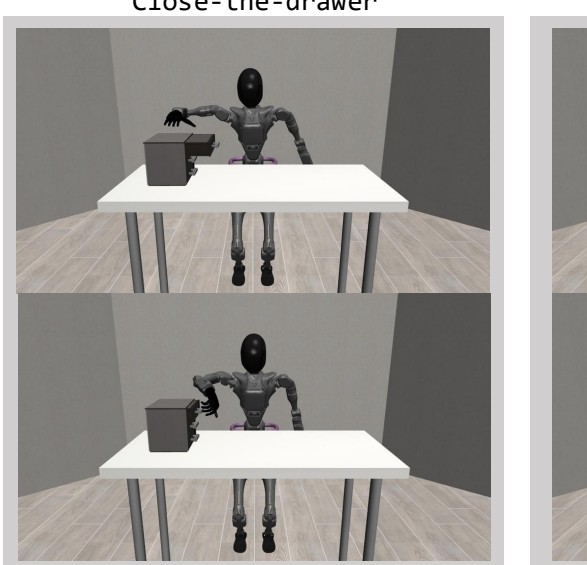
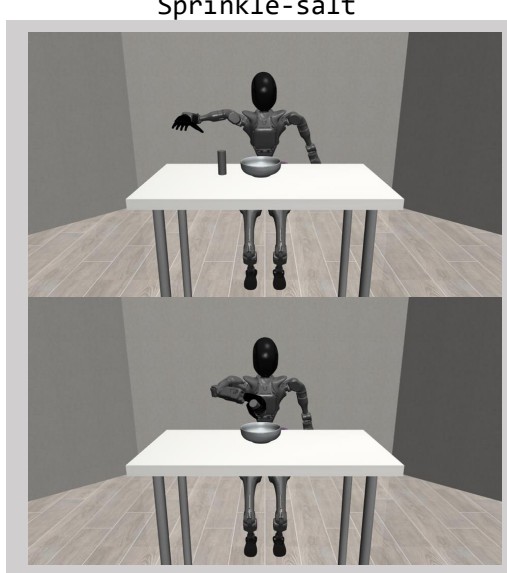

Figure 7: The screenshots of the starting and ending frames of the two simulation tasks, Close-the-drawer and Sprinkle-salt.

