# OpenReview forum: "OKAMI: Teaching Humanoid Robots Manipulation Skills through Single Video Imitation"
_robot-learning.org/CoRL/2024/Conference — CoRL 2024_

### Official Review · Reviewer_MV4q · 2024-07-07
**Useful system architecture for humanoid manipulation from single video demonstration**

**Originality:** 3
**Technical Quality:** 4
**Clarity Of Presentation:** 3
**Potential Impact:** 4
**Recommendation:** 3
**Confidence:** 5

**Review:**

Solving manipulation tasks from a single human video is a very timely and interesting problem for the robotic community. In this paper, the authors describe an end-to-end system to tackle this problem for the first time in the context of humanoid robots. I think overall the approach is sound, and the presentation is clear.

Strengths:

1. Given the use of foundation models for visual perception, the method is able to generalize across backgrounds and object instances. The authors directly demonstrate this capability in their experiments

2. A user study comparing robustness across 3 different human demonstrators is a useful experiment to show that the performance of the method holds for different users.

3. The quality of the presentation is overall high.

Weaknesses:

1. Lack of evaluation in a simulator. While real robot experiments are important, the advantage of a simulated evaluation is that it allows reproducibility of the results and direct comparison for the whole community. Given the object-centric nature of the method, using foundation models for visual understanding, it should be possible to collect human videos in the real world and test the resulting robot policy in a similar scene reproduced in simulation.

2. Trajectory warping unclear. How is trajectory warping implemented? I find the description not very clear. Does it use the same approach as in a prior work such as [5]? If yes, this should be explicitly highlighted. If not, a precise explanation of the difference between this and prior approaches should be added to the manuscript.

3. Novelty. While I think this is overall good work, it seems to me that it takes existing methods for single human video imitation from [4, 5, 6], and applies them to the new form factor of a humanoid robot. Some differentiator might be the use of foundation models to enhance visual generalization capabilities, but again this is less of an algorithmic contribution, and more of a system implementation, plugging different modules together.


Comments:

1. The description of "Reconstruct Human Motion" is too brief. A large portion of the method is hidden in the appendix, this should instead be in the main paper.

2. Bibliography: the author of [8] is wrong and the publication is missing.

**Quality Of The Limitations Section:**

3

**Questions For Rebuttal:**

During the rebuttal, the authors should address the points above. I would like to stress the fact that I believe that this work is already in the current form of high quality, and that the points I mentioned above are meant to make it exceptional.

**Robotics Focus:**

4

**Summary Of Paper:**

This work presents a method to solve humanoid robot manipulation tasks from a single human video. The approach consists of two stages. In the first stage, the provided human video is processed in order to extract a reference trajectory of the task. In the second stage, the reference trajectory is retargeted to take into account the given scene arrangement at test time, as well as the similar but different kinematics of the humanoid compared to the human demonstrator. The method is evaluated on 5 real world tasks against one baseline (ORION).

**Summary Of Recommendation:**

High quality paper, more a system architecture than an algorithmic contribution

---

### Official Review · Reviewer_MHD2 · 2024-07-19
**OKAMI Review**

**Originality:** 3
**Technical Quality:** 2
**Clarity Of Presentation:** 3
**Potential Impact:** 2
**Recommendation:** 3
**Confidence:** 3

**Review:**

This paper presents a one-shot method for imitating human demonstration videos with a humanoid robot. This is done using a two stage process that consists of trajectory extraction followed by motion re-targeting to the humanoid form factor.

It describes a robotic system designed to solve this problem, however it has several weaknesses. It does not describe the method in sufficient detail for it to be re-implemented by other researchers, and the testing of the method does not do a good job of highlighting the failure modes / weaknesses and limitations of this approach.

For example, I’m a bit unsure why the MDP formulation of the task is presented in section 3.1 as I don’t see this being made use of in any learning or optimization process.

Another example is: "This process is accomplished using a hybrid module that combines low level point clouds to identify contacts and high-level common sense reasoning to understand objects that are not directly in contact." This does not include enough information, and "common sense reasoning" is always difficult to codify.

It is also a bit unclear how object trajectories are extracted, GPT4V, IIRC can only be given static images and will probably give the names of the most prominent objects on the table, segmentation and tracking can then give consistent trajectories, are just the most moving objects used? How is change point detection done?


The English in this paper is still a bit rough in places, e.g.
L29 repetition of challenge,
L 113 methodology -> methodologies,
L248: question -> questions.

**Quality Of The Limitations Section:**

2

**Questions For Rebuttal:**

Especially the section comparing the performance of OKAMI to ORION it was not clear how grasps were computed in this case. Large differences in success rates were reported, but it was difficult to attribute these to particular failure modes and thus get a good intuition of what problems OKAMI avoids that ORION is making. In addition to this, having more than two tasks would be helpful in better demonstrating the advantages of OKAMI.

How exactly is the motion re-targeting done, given that the rendered SMPL-H model contains several relevant fingers and contact points how is the combination of these weighted and mapped to the humanoid hand and for each point how are differences e.g. in rotation and position weighted. Is this re-targeting done as an optimization procedure or a one-step calculation.

**Robotics Focus:**

3

**Summary Of Paper:**

humanoid robotic imitation by one-shot human motion retargeting from demonstration videos

**Summary Of Recommendation:**

ok tech report with missing details

---

### Official Review · Reviewer_XCSN · 2024-07-20
**OKAMI: Teaching Humanoid Robots Manipulation Skills through Single Video Imitation**

**Originality:** 4
**Technical Quality:** 4
**Clarity Of Presentation:** 4
**Potential Impact:** 4
**Recommendation:** 4
**Confidence:** 4

**Review:**

OKAMI operates through a two-staged process: first, generating a reference manipulation plan based on the human actions and object locations captured in the video, followed by retargeting these actions to the humanoid robot while adapting to real-time object positions. This approach aims to enhance the robot's ability to perform a variety of tasks, including picking, placing, pushing, and pouring, in dynamic environments.

The introduction of an object-aware retargeting method is a significant contribution to the field of humanoid robotics. By focusing on object awareness, the authors enhance the adaptability of robotic motions to varying environmental contexts. The experimental results demonstrate a notable improvement in task success rates compared to baseline methods, showcasing the effectiveness of OKAMI in generating humanoid motions that closely follow human demonstrations even in dynamically changing conditions. OKAMI's systematic generalization across spatial layouts and object instances is a noteworthy advance, positioning it well for real-world use cases.

While the tasks evaluated are diverse, more extensive experimentation could strengthen the claims made regarding generalization. Including examples of more complex manipulation tasks or environments could demonstrate broader applicability. The task success rate is a crucial metric, yet additional performance indicators, such as time taken for task completion or energy efficiency, could provide a more holistic assessment of the approach.

**Quality Of The Limitations Section:**

2

**Questions For Rebuttal:**

Can you demonstrate how the potential of OKAMI on series tasks?

**Robotics Focus:**

4

**Summary Of Paper:**

The paper presents a novel framework called OKAMI, which allows humanoid robots to imitate manipulation tasks demonstrated by humans in single RGB-D video clips. The method proposed addresses the challenges inherent in "open-world imitation from observation" settings, where robots are required to understand and adapt to new environments without explicit annotations of actions or prior knowledge of object states.

**Summary Of Recommendation:**

This is a good paper, I suggest to accept.

---

### Official Review · Reviewer_Qi9R · 2024-07-20
**Initial review**

**Originality:** 3
**Technical Quality:** 3
**Clarity Of Presentation:** 3
**Potential Impact:** 3
**Recommendation:** 3
**Confidence:** 3

**Review:**

Strengths:

The proposed method is both interesting and reasonable;

The system is deployed on a real humanoid robot;

The experiments demonstrate that it achieves competitive performance.

Weaknesses:

The technical details are insufficient. For example, it does not describe how the reference objects and target objects for each subgoal are identified and how the point clouds are extracted. Additionally, the adaptation of human trajectories to robot agents and the warping of these trajectories into the test object's coordinates are not explained.

One limitation of the work is that it does not consider object shape difference. For example, in test scenarios, the reference objects and test objects may have different sizes compared to the objects in the demonstration video. Therefore, rigidly transforming the retargeted trajectories into the coordinate of test object may lead to failure (e.g., the bottle used in test scene may be much shorter than the one in demo, but the warped trajectory try to grasp the very top of part of it).

**Quality Of The Limitations Section:**

2

**Questions For Rebuttal:**

Questions and suggestions:

Have the authors attempted tasks that require fine-grained motions, in addition to those achievable with coarse motion? Comparing the proposed method with additional baselines, such as Behavior Cloning-based approaches, could also strengthen the work;

I would suggest the authors include more implementation details on the identification of reference objects and target objects for each subgoal, the extraction of point clouds, and the adaptation and warping of human trajectories to robot agents and test object coordinates. Providing mathematical formulas would also be helpful.

**Robotics Focus:**

4

**Summary Of Paper:**

This paper addresses the problem of teaching humanoid robot manipulation skills through single human video. The proposed method identifies task relevant objects with VLM such as GPT4V, and extracts human trajectories with SMPL-H. The human trajectories are retargeted to robot agent and transformed into the local coordinates of test objects. The joint configurations for controlling the robot are then derived through inverse kinematics. Experiments are performed on a humanoid robot in the real world.

**Summary Of Recommendation:**

My rating is based on the writing issues and insufficient experiment results.

---

### Author Rebuttal · Authors · 2024-08-11

Dear reviewers and meta-reviewer,

Thanks for your time and efforts in providing us with great and constructive feedback that has strengthened our paper even further. The rebuttal file includes our revised manuscript and additional videos of a newly designed, long-horizon bimanual task.

In this rebuttal, we have revised the paper to address the reviewers’ concerns. We respond to each reviewer’s questions and concerns individually below. Specifically, we have revised the manuscript to explain the method with more details, including the parts of trajectory warping, identifying target and reference objects, and hand pose mapping. We also provide more explanation on why OKAMI succeeds while ORION fails in experiments, including comprehensive experiments in simulation.

---

### Decision · Program_Chairs · 2024-09-04

**Decision:**

Accept

**Comment:**

Strengths:
- A promising approach for teaching humanoid robots new skills from a single RGBD video demonstration, with real-world experiments.
- The paper quality is great overall. The experiments conducted are robust, with user studies and comparisons to one baseline method.


Weaknesses:
- Across the reviews, a common concern is the lack of detailed explanation on key components like trajectory warping, object identification, and adaptation processes, which affects the reproducibility and understanding of the methods.
- Reviewers raise questions around the analysis of the presented results. Why does ORION fail while OKAMI succeeds in experiments? Perhaps a more thorough analysis can be performed in simulation, including an ablation experiment.

Post rebuttal discussion:
Authors provide more details missing in initial paper, as well as analysis. Overall a good paper.